# The Hepatic Antisteatosis Effect of Xanthohumol in High-Fat Diet-Fed Rats Entails Activation of AMPK as a Possible Protective Mechanism

**DOI:** 10.3390/foods12234214

**Published:** 2023-11-22

**Authors:** Hebatallah Husseini Atteia, Nora A. AlFaris, Ghedeir M. Alshammari, Eman Alamri, Salwa Fares Ahmed, Renad Albalwi, Sahar Abdel-Latif Abdel-Sattar

**Affiliations:** 1Department of Pharmaceutical Chemistry, Faculty of Pharmacy, University of Tabuk, Tabuk P.O. Box 47512, Saudi Arabia; hatteia@ut.edu.sa; 2Department of Physical Sports Sciences, College of Sports Sciences & Physical Activity, Princess Nourah bint Abdulrahman University, Riyadh P.O. Box 84428, Saudi Arabia; naalfaris@pnu.edu.sa; 3Department of Food Science and Nutrition, College of Food and Agricultural Sciences, King Saud University, Riyadh 11451, Saudi Arabia; aghedeir@ksu.edu.sa; 4Department of Food Science and Nutrition, University of Tabuk, Tabuk P.O. Box 47512, Saudi Arabia; ralbawi23456@hotmail.com; 5Department of Anatomy, Faculty of Medicine, University of Tabuk, Tabuk P.O. Box 47512, Saudi Arabia; s.hasan@ut.edu.sa; 6Department of Histology, Faculty of Medicine, Assiut University, Assiut 71515, Egypt; 7Applied College, University of Tabuk, Tabuk P.O. Box 45712, Saudi Arabia; sabdelgelil@ut.edu.sa

**Keywords:** xanthohumol, non-alcoholic fatty liver, high-fat diet, steatosis, oxidative stress, inflammation, rats

## Abstract

Obesity is the leading cause of non-alcoholic fatty liver disease by provoking hyperglycemia, hyperlipidemia, insulin resistance, oxidative stress, and inflammation. Low activity of AMP-activated protein kinase (AMPK) is linked to obesity, liver injury, and NAFLD. This study involves examining if the anti-steatosis effect of Xanthohumol (XH) in high-fat diet (HFD)-fed rats involves the regulation of AMPK. Adult male rats were divided into five groups (*n* = 8 each) as control (3.85 kcal/g); XH (control diet + 20 mg/kg), HFD (4.73 kcl/g), HFD + XH (20 mg/kg), and HFD + XH (30 mg/kg) + compound c (cc) (0.2 mg/kg). All treatments were conducted for 12 weeks. Treatment with XH attenuated the gain in body weight, fat pads, fasting glucose, and insulin in HFD rats. It also lowered serum leptin and free fatty acids (FFAs) and improved glucose and insulin tolerances in these rats. It also attenuated the increase in serum livers of liver marker enzymes and reduced serum and hepatic levels of triglycerides (TGs), cholesterol (CHOL), FFAs, as well as serum levels of low-density lipoproteins cholesterol (LDL-c) oxidized LDL-c. XH also reduced hepatic levels of malondialdehyde (MDA), nuclear accumulation of NF-κB, and the levels of tumor necrosis-factor-α (TNF-α) and interleukin-6 (IL-6) while stimulating the nuclear levels of Nrf2 and total levels of glutathione (GSH), superoxide dismutase (SOD), and catalase (CAT) in these HFD-fed rats. At the molecular levels, XH increased hepatic mRNA expression and phosphorylation of AMPK (Thr^72^) and reduced the expression of lipogenic genes SREBP1c and ACC-1. In concomitance, XH reduced hepatic liver droplet accumulation, reduced the number of apoptotic nuclei, and improved the structures of nuclei, mitochondria, and rough endoplasmic reticulum. Co-treatment with CC, an AMPK inhibitor, completely abolished all these effects of XH. In conclusion, XH attenuates obesity and HFD-mediated hepatic steatosis by activating hepatic AMPK.

## 1. Introduction

During the last decade, modernization and the sedentary lifestyle associated with consuming a Western diet rich in high fats has disturbed our energy balance and increased the prevalence of type-2 diabetes mellitus (T2DM) and metabolic syndrome (MetS) [1]. Nonalcoholic fatty liver disease (NAFLD) is the most common manifestation of T2DM and metabolic syndrome (MetS); it is induced by the associated insulin resistance (IR), hyperglycemia, and hyperlipidemia [2]. While the disease is initially established as simple steatosis, it can progress to advanced stages such as hepatic steatosis and non-alcoholic steatosis (NASH) due to second hits such as inflammation and oxidative stress, which are currently considered major molecular pathways underlying NAFLD [3].

However, despite the great success in drug discovery and enrolling them in clinical practice, until now, no definite therapy is available to treat NAFLD, which can be explained by the complexity of the disease, the existence of several pathological pathways, and the inability of a particular drug to act on all these pathways [4]. Indeed, NAFLD is a complex disorder associated with alterations in numerous signaling pathways involved in inflammation, oxidative stress, and lipogenesis [5]. Within this view, NAFLD is associated with the activation of the nuclear factor-kappa beta (NF-κB), a master inflammatory mediator [6], as well as the suppression of the nuclear factor erythroid factor-2 (Nrf2), a major antioxidant transcription factor [7]. In addition, NAFLD is associated with sustained activation of lipid synthesis transcription factors such as the regulatory element-binding proteins-1/2 (SREBP1/2) that stimulated triglycerides (TGs) and cholesterol (CHOL) synthesis, respectively [8]. However, these studies have also shown that the activation of Nrf2 or suppression of NF-κB and/or SREBP1/2 alleviated hepatic steatosis and protected against the development of NASH and hepatic fibrosis in experimental animals. Therefore, it is reasonable that any drug targeting all these pathways will provide a novel insight into the treatment of NAFLD.

Research continues and recent studies have listed the 5-AMP-activated protein kinase (AMPK) as a novel regulator of MetS and NAFLD [9]. AMPK is the major regulator of metabolism that can regulate food intake, whole-body expenditure, and glucose and lipid hemostasis by acting on different tissues, such as the hypothalamus, liver, pancreas, muscles, and adipose tissue [9]. AMPK is activated mainly by energy depletion (low ATP) and glucose deprivation to stimulate the production of ATP by promoting catabolic pathways while suppressing anabolic pathways [10]. Experimental and clinical studies conducted in animals or patients with NAFLD have shown a significant reduction in activities of AMPK in the majority of tissues, which were also correlated to the degree of hyperglycemia, IR, and hepatic steatosis [11]. Interestingly, pharmacological activation of AMPK alone, by AICAR, metformin, and other natural drugs, attenuated NAFLD symptoms and improved liver function and structure by alleviating oxidative stress, inflammation, and apoptosis [11,12,13]. In particular, these effects were attributed to the ability of AMPK to act as an upstream regulator to reduce food intake and stimulate peripheral glucose uptake and the expression of glucose transporters (GLUT-4) [14]. In addition, AMPK stimulated hepatic FAs oxidation and peripheral insulin sensitivity and inhibited hepatic gluconeogenesis, de novo lipogenesis, and adipose tissue lipolysis by acting on several targets such as acetyl-CoA carboxylase (ACC), fatty acid synthase, SREBP1/2, AS160, the transducer of regulated CREB activity 2 (TORC2), PPARγ, and other molecules [9,11,15]. Furthermore, AMPK can directly alleviate oxidative stress, inflammation, and fibrosis in the liver of NAFLD animals by activating Nrf2/antioxidant axis, improving mitochondria biogenesis, and inhibiting NF-κB and Ca+2 signaling [16]. Therefore, searching for safe drugs that act on AMPK is an urgent need to treat NAFLD.

Xanthohumol (XH), C_21_H_22_O_5_, is the most active flavonoid in the hop plant, which has been recently described as a unique anti-aging molecule [17]. XH accounts for about 0.1–1% of female inflorescences of the hop plant and is secreted as resin in the trichomes on the underside of young leaves [18]. XH can be extracted from plants or can be synthesized commercially. During the last decades, several studies were conducted to understand the pharmacological properties of XH. Currently, it is well-established that XH exerts potent anti-obesity, anti-diabetic, hypoglycemic, and hypolipidemic effects of XH [19,20,21,22,23,24,25,26]. Indeed, XH was able to attenuate obesity by suppressing the differentiation of preadipocytes through inhibiting CCAAT enhancer-binding proteins (C/EBP) and the peroxisome proliferator-activated receptor (PPAR), as well as promoting mature adipocyte apoptosis [27,28,29,30,31]. XH was also able to attenuate body weight gain in obese animals by inhibiting fat absorption in the intestine [32]. Furthermore, XH can alleviate hyperglycemia in both T1DM and T2DM animal models. Within this view, mechanisms of action include suppressing intestinal glucose absorption (i.e., α-glucosidase) and improving peripheral glucose sensitivity [24,33]. XH can also attenuate the increments in CHOL and TGS, low-density lipoproteins (LDL), and increase the ratio of high-density lipoprotein (HDL) to LDL by acting through several mechanisms, such as decreasing hepatic apolipoprotein B (ApoB) release, decreasing fee fatty acid influx form adipose tissue, and inhibiting diacylglycerol acyltransferase and cholesteryl ester transfer protein (CETP) [21,34,35].

In addition to its metabolic effect, the multi-organ protective function of XH was described in a variety of chronic renal, cardiac, hepatic, and neural disorders and was attributed to its independent antioxidant and anti-inflammatory effect, mainly due to regulating AMPK, Nrf2, and NF-κB signaling [36,37,38,39]. Indeed, XH can alleviate oxidative stress and inflammation by activating Nrf2 and inhibiting NF-κB [40,41]. In diabetic animal models, XH prevented cognitive impairment and nephropathy by activating Nrf2 and suppressing NF-κB [42,43]. Interestingly, XH stimulated lipolysis and inhibited adipogenesis in cultured 3T3-L1 adipocytes and primary human subcutaneous preadipocytes by activating AMPK [44]. It also alleviated diabetic oxidative stress and accelerated wound healing in streptozotocin (STZ)-insulin-deficient diabetic rats by activating AMPK/Nrf2 signaling [45]. Furthermore, the anti-angiogenic and anti-tumor effects of XH on endothelial cells were mediated by AMPK-induced deactivation of endothelial nitric oxide (eNOS) [46]. It also prevented lipopolysaccharides (LPS) induced lung injury by stimulating the AMPK/Nrf2 antioxidant axis [47].

Interestingly XH is also able to attenuate lipogenesis in cultured hepatocytes and protect against NAFLD in mice and rats by suppressing the hepatic SREBP1 and diacylglycerol acyltransferase (DAG), activation of farnesoid X receptor (FXR), antagonizing PPARγ, and stimulating the Nrf2/RAGE/NF-κB axis [48,49,50,51]. This data indicates that XH may act by various mechanisms under a common umbrella. Given the role of AMPK in regulating all these pathways, we are interested in examining if XH hepatic and anti-metabolic protective effects are mainly mediated by targeting AMPK.

Therefore, in this study, we tested the hypothesis that chronic co-treatment with XH could alleviate NAFLD in high-fat-diet-fed rats by activating AMPK, which acts as an upstream mechanism to suppress lipogenesis, oxidative stress, and inflammation.

## 2. Materials and Methods

### 2.1. Animals

Wistar-type male Wistar–Dawley rats weighing 150 ± 15 g and aged 8 weeks were used in this study and were provided by the animal house at King Saud University, Saudi Arabia. During all experimental conditions, all animals were housed in plastic cages (four rats/cage) with a stable room temperature of 21 ± 1 °C and a 12 h dark/light cycle. The experimental designs, treatment, tissue collection, measurements, and analysis were approved by the animal care and use committee at King Suad University (IRB # KSU-SE-21-34), which follows the animal ethics guidelines of the US National Institutes of Health (NIH publication No. 85-23, revised 1996).

### 2.2. Drugs and Diets

Xanthohumol (XH) from hop (*Humulus lupulus*) (formula = C_21_H_22_O_5_/cat # X0379) was purchased from Sigma Aldrich, St Louis, MO, USA). Dorsomophin, also known as compound C (CC), a selective AMPK inhibitor) (cat # 171260) was also provided from Sigma Aldrich. XH and CC were always freshly prepared in DMSO. The final concentration of dimethylsulfoxide (DMSO) in all preparations was 0.1%. The control diet (cat # D12450J; 3.85 kcal/g) (20% protein, 70% carbohydrate, and 10% fat [lard = 20 g%; 180 g]) and HFD (cat # D12451; 4.73 kcl/g) (20% protein, 35% carbohydrate, and 45% fat [lard = 177 g%; 1598 kcal]) were purchased from Research Diets, NJ, USA. These diets are regularly used in our laboratories to induce HFD in Wistar and Sprague rats, and their composition is shown in some previous studies and on the provider’s website [52,53].

### 2.3. Experimental Designs and Groups

A total of 40 rats were included in this study. The animals were selected randomly after a 1-week adaptation and then segregated into 5 groups (n = 8 rats/group) as follows: (1) control group: fed only the control diet and co-treated orally with 0.1% DMSO as a vehicle; (2) control + XH-treated group: fed the control diet and co-treated orally with XH at a dose of 20 mg/kg; (3) HFD-fed group: fed HFD and co-treated orally with 0.1% DMSO; (4) HFD + XH-treated group: fed HFD and concomitantly and orally treated with XH (20 mg/kg); and (5) HFD + XH + CC-treated group: fed HFD, orally administered XH (20 mg/kg); and i.p. treated with CC at a dose of (0.2 mg/kg). Diets and all treatments were given daily for a total period of 12 weeks [51,54]. Body weights and food intake were measured every week.

The dose of XH was also based on studies of others, which have shown weight-reducing, hypoglycaemic, hypolipidemic, hepatic protective, neuroprotective, hepatic antioxidant, and anti-inflammatory effects in mice and rats [40,51,55]. CC was used in vivo to block the hepatic AMPK activation in rats [24,56].

### 2.4. Oral Glucose and Intraperitoneal Insulin Tolerance Test (OGTT and IPITT)

During the last week of the study, all 12-h fasted animals were orally administered a 2 g/kg glucose solution. Two days later, they were administered insulin (2 units/kg). In both tests, blood samples were collected in Na-fluoride-containing tubes from each rat at different time intervals (0.0, 15, 30, 60, and 120 min). Plasma samples were collected from all rats after cardiac puncture and centrifugation (500× *g*/10 min/room temperature). Plasma samples were collected at 0.0. min during the OGTT, and were used as basal levels of glucose and insulin, which were measured using special ELISA kits (cat #81695, Chrystal Chem, IL, USA, and cat # ERINS, ThermoFisher, Waltham, MA, USA). The degree of IR as measured by the homeostasis model of insulin resistance (HOMA-IR) was calculated based on others [57], who used the following equation: {HOMA-IR = [fasting glucose (mg/dL) × fasting insulin (µIU/mL)]/405. All measurements were performed using a plate reader (model # ER-181s, New Delhi, India)

### 2.5. Collection of Serum, Tissues, and WAT Fat Pads

Stools of all groups of rats were collected during the last two weeks of the experiment, dried, and kept at −20 °C until use. By the end of the experimental protocol, the rats of all groups were anesthetized using a ketamine/xylazine mix (80/10 *v*/*v*/kg). Once anesthesia was confirmed, blood samples were directly collected from the heart into gel-contained tubes, centrifuged (500× *g*/10 min/room temperature), and serum samples were collected. These samples were further maintained in the fridge and used freshly to measure the levels of some biochemical markers. Then, all rats were sacrificed by cervical dislocation, and the livers were carefully removed, put on ice, washed with ice-cold normal saline to remove excess blood, and weighed. The liver lobules collected from different areas of each liver were cut into smaller sections, which were randomly selected and either placed in 10% buffered formalin (for histological evaluation) or snap frozen at −80 °C, which were used later for further various experiments. In addition, post-mortem, white adipose tissue (WAT), including the subcutaneous, epididymal, and retroperitoneal (perirenal), was collected and weighed as a WAT fat pad.

### 2.6. Liver Tissue Processing

The extraction of lipids from the liver samples was performed based on the method of Folch et al. [58]. The nuclear and cytoplasmic extracts were separated from selected frozen liver tissues using a commercially available extraction kit (Cat. No. 4110147; Bio-Rad, CA, USA). Total RNAs were extracted from other liver tissues using a commercial kit (Cat. # 74004; Qiagen, Germany). The preparation of the tissue homogenates for the biochemical analysis was conducted using ice-phosphate buffered saline with a pH of 7. The tissue protein extract was prepared for western blotting by homogenizing the tissues in radioimmunoprecipitation (RIPA buffer). In all cases, homogenate supernatants were collected after centrifugation of the samples at 12,000× *g* for 10 min at 4 °C. Once needed, protein levels were determined in any given sample using the BCA Protein Assay Kit (Cat # 23225, ThermoFisher, USA). All samples containing homogenates, RNA, and proteins were preserved at −80 °C for further processing.

### 2.7. Analyses of Lipids in the Liver and Serum

Serum, fecal, and liver levels of total CHOL and total TGs were measured using rat’s specific CHOL and TGs assay kits (Cat # 10009582, Cayman Chemicals, Ann Arbor, MI, USA), (Cat # ECCH-100, BioAssay Systems, Hayward, CA, USA). Serum and hepatic levels of LDL-c and HDL-c were measured using rat’s specific LDL-c and HDL-c determination kits (Cat # 79960 and Cat # 79970, Crystal Chemicals, Elk Grove Village, IL, USA). Serum and hepatic levels of free fatty acids (FFAs) were determined using rat’s specific FFA ELISA determination kit (Cat # MBS014345, MyBioSource, San Diego, CA, USA). All measurements were performed using a plate reader (model # ER-181s, New Delhi, India). All protocols were performed following the manufacturer’s instructions and for eight rats/groups.

### 2.8. Biochemical Analysis in the Serum Samples

Serum leptin levels were measured using a rat’s specific leptin ELISA kit (Cat # ab100773, Abcam, Cambridge, UK). Serum levels of aspartate aminotransferase (AST) were measured using an AST determination ELISA kit for rats (Cat # MBS264975; MyBioSorces, CA, USA; respectively). A rat’s specific GGT ELISA kit was used to measure the serum levels of gamma-glutamyl transpeptidase (GGT) (Cat # MBS269614; MyBioSorces, San Diego, CA, USA). Alanine aminotransferase (ALT) levels were measured using an ELISA kit (Cat # MBS264975; MyBioSorces, San Diego, CA, USA). All analyses were conducted for a total number of rats of 8/group and as instructed by each provided kit. IL-6 and TNF-α ELISA kits were used to measure serum and hepatic homogenate levels of tumor necrosis factor-alpha (TNF-α) and interleukine-6 (IL-6) (Cat # RTFI01177 and Cat # RTEB1811, Assay Genie, London, UK).

### 2.9. Biochemical Analysis in the Liver Homogenates

SOD ELISA kit was used to measure the levels of total superoxide dismutase (SOD) in the liver homogenates (Cat # RTFI00215, Assay Genie, London, UK). The liver homogenate levels of lipid peroxides, as indicated by the total malondialdehyde (MDA) levels, were measured in an MDA ELISA kit (Cat # MBS268427, MyBioSource, CA, USA). The total levels of reduced glutathione (GSH) and glutathione peroxidase (GPX) were measured using GSH and GPX rat-specific ELISA kits (Cat # RTEB0206 and Cat # RTEB1811, Assay Genie, London, UK). Rat’s specific CAT ELISA kit was used to measure total levels of catalase (CAT) in the liver homogenates (Ca # MBS726781, MyBioSource, CA, USA). The cytoplasmic and nuclear levels of the Nrf2 and NF-κB in the cytoplasm and the nuclear extracts were measured using the Nrf2, and fractions were assessed by rats special ELISA kits (Cat # MBS752046 and Cat MBS453975, MyBiosources, CA, USA). All measurements were performed using a plate reader (model # ER-181s, New Delhi, India). All protocols followed the instructions of the kits and were done for *n* = 8 samples/group.

### 2.10. Real-Time Polymerase Chain Reaction (q-PCR)

The primer pair sequences used for the q-PCR reaction are shown in Table 1. Total RNA was extracted using a commercial kit (cat 74004; Qiagen, Germany). The first-strand cDNA was synthesized using a cDNA synthesis commercial kit (cat K1621, ThromoFisher, Waltham, MA, USA). The amplification reaction was performed in a CFX96 PCR machine using the Ssofast Evergreen Supermix kit as instructed by the manufacturer (cat 172-5200, BioRad, Hercules, CA, USA). The amplification reactions were set as follows: (1) heating (1 cycle/98 °C/30 s), (2) denaturation (40 cycles/98 °C/5 s), (3) annealing (40 cycles/60 °C/5 s), and (4) melting (1 cycle/95 °C/5 s/step). The relative mRNA expression of all target genes was presented after the normalization of β-actin using the 2ΔΔCT method.

### 2.11. Western Blotting

The procedure for western blotting was conducted as routinely performed in our laboratories. In brief, total proteins extracted from all livers were diluted and prepared in the loading dye (3 µg/µL) and then separated by the Sodium Dodecyl Sulphate-Polyacrylamide Gel Electrophoresis (SDS PAGE) (30 µg/well). The protein was then blocked with skimmed milk and incubated with the primary antibody against AMPKα (Cat # 5831, 62 kDa, 1:1000, Cell signaling Technology, Waltham, MA, USA), p-AMPKα (Thr172) (Cat. No. 50081, 62 kDa, 1:500, Cell signaling Technology, USA), and β-Actin Antibody (Cat. No. sc-69879, 43 kDa:1:2000, Santa Cruz, CA, USA). Then, membranes were washed again and incubated with each corresponding antibody. Membranes were stripped up to three times, and internal control was used for normalization. Band detection was performed using a chemiluminescence detection kit (Cat # 11520709001, Roche Diagnostic, Mannheim, Germany). All gels were scanned and analyzed using the C-DiGit blot scanner (LI-COR, USA) and associated software. The protein expression of each target was presented relative to the expression of its β-actin.

### 2.12. Histological Studies

Liver tissues were fixed in 10% buffered formalin for 24 h and then were embedded in paraffin. Then, they were sectioned using a rotatory microtome at a thickness of 5-µ and routinely stained with hematoxylin and eosin (H&E) for pathological evaluation. All images were captured using a light microscope with a camera (Eclipse E800, Nikon, Tokyo, Japan) under 200× power.

### 2.13. Statistical Analysis

Statistical analysis for all measured parameters was conducted using the GraphPad Prism statistical software package (version 8) and the 1-way ANOVA test. Normality was tested using the Kolmogorov–Smirnov test, and Tukey’s test was used as a post hoc test. Data were presented as mean ± SD. The value of significance was set at *p* < 0.05.

## 3. Results

### 3.1. XH Reverses the Gain in Food Intake and Body Weight in HFD-Fed Rats

Body weight and food intake were progressively increased through the 12 weeks of the study in all groups of rats (Figure 1A–D). No significant variations in body weight gain and food intake were seen in the control (STD-fed) and XH-treated rats (Figure 1A–D). Body weight and food intake were significantly increased in HFD-fed rats as compared to the STD group, both of which were reduced to basal levels in HFD-fed rats (Figure 1A–D). No significant variations in body weight gain and food intake were seen between the HFD-fed rats and HFD + XH + CC-treated rats (Figure 1A–D).

### 3.2. XH Improves Glucose Levels after Glucose and Insulin Tolerance Tests in HFD-Fed Rats

The alterations in the levels of glucose over 120 min post the OGTT and IPITT in all groups of rats are shown in Figure 2A–F. After 30 and 60 min, glucose levels and the percentages of increase in glucose levels were significantly reduced post-glucose or insulin administration, respectively, in XH-fed rats as compared to STD rats (Figure 2A,B,D,E). As a result, the area under the curve (AUC) corresponding to the increment in glucose levels post-OGTT was significantly lower, and the percentages of reduction in glucose levels post-OGTT were higher in XH-fed rats as compared to STD-fed rats (Figure 2C,F). The glucose levels measured during the OGTT and IPITT were significantly higher at 0.0, 30, 60, 90, and 120 min in HFD-fed rats as compared to STD-fed and XH-treated rats (Figure 2A,D). In addition, the percentages of gain in glucose levels as measured 30, 60, 90, and 120 min after glucose administration, as well as the calculated AUC, were significantly higher in HFD-fed rats as compared to STD and XH-treated rats (Figure 2B,C). In the same manner, the percentages of reduction in glucose levels in glucose levels as measured 30, 60, 90, and 120 min after insulin administration, as well as their calculated AUC, were significantly lower in HFD-fed rats as compared to STD and XH-treated rats (Figure 2E,F). These changes in glucose levels, percentages of gain/reduction in glucose levels, and their areas AUC observed in HFD-fed rats were reversed in HFD + XH–treated rats compared to HFD-fed rats (Figure 2A–F). A typical mirror image for the changes in glucose levels after glucose and insulin administration was seen in HFD + XH + CC as compared to HFD-fed rats (Figure 2A–F).

### 3.3. XH Attenuates Markers of Adiposity in HFD-Fed Rats

Among all measured parameters, XH-treated rats showed significantly lower levels of glucose, HOMA-IR, serum FFAs, and serum TNF-α as compared to STD-fed rats (Table 2). Final body weights and the weights of subcutaneous, epididymal, and peritoneal fats, as well as fasting glucose levels, fasting insulin levels, HOMA-IR and serum leptin, TNF-α, and FFAs, were significantly increased in HFD-fed rats as compared to STD-fed rats (Table 2). The levels of all these biochemical endpoints were significantly reduced in HFD + XH-treated rats when compared to HFD-fed rats and were increased again in HFD + XH-treated rats as compared to HFD + XH-treated rats (Table 2). No significant variations in the levels of all these markers were seen between HFD-fed rats and HFD + XH + CC-treated rats (Table 2).

### 3.4. XH Ameliorates Dyslipidemia and the Increase in Hepatic Lipid Levels in HFD-Fed Rats

Serum and hepatic levels of TGs, CHOL, and FFAs, as well as serum levels of LDL-c and ox-LDL-c, were significantly reduced in XH-treated rats as compared to STD-fed rats (Table 3). In addition, the hepatic levels of HDL-c were significantly increased, whereas the fecal levels of TGs and CHOL were not statistically varied in XH-treated rats when compared to STD-fed rats (Table 3). Except for HDL-c, which was reduced, the serum and hepatic levels of TGs, CHOL, and FFAs, as well as serum levels of LDL-c and ox-LDL-c, and the fecal levels of CHOL and TGs were significantly increased in HFD-fed rats as compared to STD-fed rats (Table 3). On the contrary, HFD-fed rats showed significantly lower serum and hepatic levels of TGs, CHOL, and FFAs, as well as reduced serum levels of LDL-c and ox-LDL-c as compared to HFD-fed rats (Table 3). However, HDL-c levels were significantly increased in HFD + XH-treated rats as compared to HFD-fed rats (Table 3). These effects were reversed in HFD + XH + CC-treated rats as compared to HFD + XH-treated rats (Table 3). In addition, no significant variations in the serum or hepatic levels of all these lipids were seen between the HFD-fed and HFD + XH + CC-treated rats (Table 2). Of note, fecal levels of CHOL and TGs were not significantly altered between HFD-fed, HFD + XH, and HFD + XH + CC-treated rats when compared with each other (Table 3).

### 3.5. XH Reduced Liver Damage Marker Enzymes in All Groups of Rats

No significant differences in ALT, AST, and GTT serum levels were seen between the STD-fed and XH-treated rats (Table 4). Serum levels of all these hepatic marker enzymes were significantly increased in HFD-fed rats as compared to STD and XH-treated rats (Table 4). Serum levels of ALT, AST, and GTT were significantly lower in HFD + XH-treated rats as compared to HFD-fed rats (Table 4). No variations in ALT, AST, and GTT levels were seen between the HFD + XH and HFD + XH + CC-treated rats (Table 4).

### 3.6. XH Attenuates Lipid Peroxidation and Stimulates Nrf2 Activities and Antioxidant Levels in the Livers of HFD-Fed Rats

Levels of MDA were significantly reduced, whereas the cytoplasmic and nuclear levels of Nrf2, as well as levels of GSH, SOD, and CAT, were increased in the livers of HFD-fed rats as compared to STD-fed rats (Figure 3A–D). Levels of MDA were significantly increased, whereas the cytoplasmic and nuclear levels of Nrf2, as well as levels of SOD and GSH, and CAT, were reduced in the livers of HFD-fed rats as compared to HFD-fed rats (Figure 3A–D). Livers of the HFD + XH-treated rats showed significantly lower levels of MDA and higher levels of GSH, SOD, and CAT and increased levels of cytoplasmic and nuclear levels of Nrf2 as compared to HFD-fed rats. All these events were reversed in the livers of HFD + XH + CC-treated rats compared to HFD + XH-treated rats, which were not significantly different from those measured in HFD-fed rats (Figure 3A–D).

### 3.7. XH Inhibits the Activation of NF-κB and the Production of Inflammatory Cytokines in the Liver of HFD-Fed Rats

The cytoplasmic and nuclear levels of NF-κB and the levels of TNF-α and IL-6 were significantly increased in the livers of HFD-fed rats as compared to STD-fed rats (Figure 4A–C). The cytoplasmic and nuclear levels of NF-κB and the levels of TNF-α and IL-6 were significantly reduced in the livers of both the XH-treated and HFD + XH-treated rats as compared to either the STD or HFD-fed rats, respectively (Figure 4A–C). The cytoplasmic and nuclear levels of NF-κB and the levels of TNF-α and IL-6 were increased in the livers of HFD + XH + CC-treated rats as compared to HFD + XH-treated rats (Figure 4A–C). No significant differences in the levels of all these markers were seen when HFD-fed rats were compared with HFD + XH + CC-treated rats (Figure 4A-C)..

### 3.8. XH Suppresses the Phosphorylation (Activation) of AMPK, ACC-1, and SREBP1 in the Livers of HFD-Fed Rats

Total phosphorylation levels of AMPK, as well as mRNA levels of ACC-1 and SREBP1, were measured in the livers of all groups of rats (Figure 5A–C). The ratio of activation was calculated as percentages of phosphorylated protein divided by the total levels of that protein, all normalized to β-actin. Hepatic levels of total protein of AMPK were not significantly varied between all groups of rats (Figure 5A). The mRNA and phosphorylation of AMPK (Thr92) (activity) were significantly decreased, whereas mRNA levels of ACC-1 and SREBP1 were increased in the livers of HFD-fed rats as compared to STD and XH-fed rats (Figure 5A–C). The mRNA and protein phosphorylation of AMPK were significantly decreased, whereas mRNA levels of ACC-1 and SREBP1 were significantly decreased in the livers of XH-treated and HFD + XH-treated rats as compared to STD or HFD-fed rats, respectively (Figure 5A–C). Opposing this, the mRNA and rate of protein phosphorylation levels of AMPK were significantly decreased, whereas mRNA levels of SREBP1 and ACC-1 were increased in the livers of HFD + XH + CC-treated rats as compared to HFD + XH-treated rats (Figure 5A–C). No significant variations in the levels of all these biochemical endpoints were seen between the HFD-fed and HFD + XH + CC-treated rats (Figure 5A–C).

### 3.9. XH Improves Liver Histology and Ultrastructures in HFD-Fed Rats

The livers of the control and XH-treated rats showed normal histological features, including central vein, hepatocytes, and sinusoids (Figure 6A,B, respectively). At the ultrastructural levels, control and XH-treated rats also showed normal hepatocytes with intact nuclei surrounded by intact envelopes, chromatin, rough endoplasmic reticulum (RER), and mitochondria (Figure 7a,b, respectively). On the contrary, the livers of HFD-fed rats showed an increased number of cytoplasmic fat vacuolization, which was dominant over the whole fields (Figure 6C). In addition, the livers of this group of rats showed dilated sinusoids and an increased number of invading immune cells and apoptotic pyknotic nuclei (Figure 6C). The electron microscopy images of the livers of HFD-fed rats also showed obvious apoptotic and degenerated hepatocytes with pyknotic nucleus, increased amounts of lipids droplets and myelin figures, and damaged and dilated RER (Figure 7c). The livers of HFD + XH-treated rats showed much improvement in the liver structure with normally appeared hepatocytes, and sinusoids were seen in the livers (Figure 6D). However, the livers of these rats remained to show some cytoplasmic lipid vacuolization in the areas distal to the central vein (Figure 6D). Their ultrastructural images also showed an improvement in the hepatocyte and bile canaliculus structure with partial changes in some mitochondria and the RER (Figure 7d). A typical picture with the same histological and ultrastructural changes with increased nuclear damage and several fat droplets that were seen in the HFD-fed rats was also seen in the livers of HFD + XH + CC-treated rats (Figure 6E and Figure 7e).

## 4. Discussion

Previous studies have shown a protective effect of XH against obesity and NAFLD in rodents [32,50]. Unfortunately, the upstream mechanism responsible for these effects is still unclear. In this study, we are supporting these findings and further extending them. The overall conclusion obtained from the data of this study confirms that the anti-obesity and anti-steatosis protective effect of XH involves potent activation of AMPK in both the liver and WAT of HFD-fed rats. Such XH-dependent activation of AMPK is indispensable to enhance WAT energy expenditure, suppress hepatic de novo lipogenesis, and attenuate hepatic inflammation, oxidative stress, fibrosis, and apoptosis via direct regulation of several transcription factors and enzymes such as thermogenesis-related genes, ACC-1, SREBP1, Nrf2, and NF-κB. However, suppressing AMPK by CC completely abolished all the metabolic and hepatic beneficial effects of XH in these HFD-fed rats, which showed similar phenotypes and the biochemical picture seen in untreated model HFD-fed rats. A schematic diagram showing this mechanism of action of XH is shown in Figure 8.

Obesity and IR are the major risk factors triggering NAFLD by disturbing adipose tissue lipogenesis and increasing the hepatic influx of inflammatory cytokines, FFAs, and adipokines from the impaired WAT [59]. The first observation obtained from this study was the obvious anti-obesity, hypoglycemic, and peripheral insulin-improving effects of XH, which were independent of modulating food intake. These effects have been also shown in obese and diabetic individuals and humans [25,60,61,62]. In addition, accumulating data has examined some mechanisms behind the anti-obesity and anti-diabetic effects of XH in HFD and diabetic animal models. In this regard, some authors have found that XH reduces fasting hyperglycemia through its ability to increase intestinal fat absorption and reduce feeding efficiency [63]. However, this was not supported by our data. XH failed to alter TGs and CHOL levels in the feces, as well as change food intake in control rats. However, the reduction in food intake in HFD-fed animals could be related to the suppressor effect of XH on leptin release from adipose tissue as a result of reducing fat mass. Indeed, circulatory leptin levels positively correlate with fat mass and body mass index (BMI) [63]. On the other hand, other authors have shown that XH can attenuate adipogenesis by inhibiting adipocyte differentiation/number and stimulating energy expenditure, mitochondria uncoupling, and thermogenesis [27,28,40]. Despite this, the precise mechanism for these anti-adipogenic effects is still lacking.

Herein, we further extend these findings and show that the anti-obesity and hypoglycemic effects of XH depend mainly on the activation of AMPK, which is considered a valuable finding and has never been shown before in vivo. Indeed, treatment with CC prevented the reduction in body weight, fat pad weights, and serum levels of TNF-α, FFAs, and leptin in HFD-fed rats that received XH. Supporting our data, the previously reported in vitro study of Samuels et al. [44] has shown that XH strongly inhibits differentiation and stimulates thermogenesis and browning in cultured 3T3-L1 adipocytes by activating AMPK. Our data also support many other studies in rodents with obesity, metabolic syndrome, and T2DM, where the activation of AMPK has also shown an anti-adiposity effect [52,56,64,65]. In addition, activation of AMPK by exercise, AMPK activators (e.g., metformin and a-769662), and other natural flavonoids (e.g., berberine) attenuated obesity, fat deposition, and IR in HFD and T2DM [64,65,66]. In addition, AMPK is indispensable for improving insulin sensitivity via increasing the translocation of glucose transporters [67]. Furthermore, AMPK reduces adipogenesis by stimulating the expression of several adipogenic genes, such as C/EBPβ, PPARγ, C/EBPα, FAS, aP2, and SREBP-1c via the activation of the WNT/β-catenin pathway [66,68,69]. Moreover, AMPK inhibits adipose tissue lipolysis and adipogenesis by suppressing adipocyte differentiation, increasing the phosphorylation of the hormone-sensitive lipase (HSL) at Ser 565, and stimulating energy expenditure, thermogenesis, and mitochondria uncoupling [64,70,71,72]. In addition, AMPK induces adipose tissue browning via other mechanisms, including decreasing the DNA methylation of the PRDM16 promotor [73]. Therefore, at this stage, we could postulate that the activation of AMPK is a major mechanism underlying the anti-adipogenic, thermogenic, insulin sensitivity improvement, and anti-obesity effect of XH. Unfortunately, we could not measure the levels of activation of AMPK in this study, which is considered a limitation. This could be targeted in future studies to confirm these data.

Hepatic lipotoxicity and dyslipidemia are major contributing factors to the development of hepatic steatosis and the progression to NASH [74]. The increased dietary intake of FFA, the higher influx of FFA, and hyperglycemia, as well as hepatic oxidative stress and inflammation, are closely related to hepatic steatosis and dyslipidemia [8,75]. Drugs that can reduce hepatic de novo lipogenesis and/or stimulate FA oxidation are protected against the development of NAFLD and the progression to NASH in experimental animals and humans [76,77]. The data of this study also confirms an exceptional and potent hypolipidemic effect of XH not only in HFD-fed rats but also in control rats, which were treated with this compound. These effects were associated with reducing hepatic and serum levels of FFAs, TGs, and CHOL that are concomitant with a reduction in serum levels of LDL-c and HDL-c, as well as in the hepatic levels of SREBP1a. These data follow many other previous studies that have shown similar effects of XH in human and animal models of obesity and NAFLD [32,40,48,50]. Indeed, XH could ameliorate hepatic steatosis and lipogenesis adipose tissue-independent mechanism that involves decreasing apolipoprotein B (apo B) secretion, downregulating/inhibiting SRREBP1/2, activating the farnesoid X receptor (FXR), suppressing diacylglycerol acyltransferase (DGAT) activity, increasing the fecal release of fats, and antioxidant-mediated suppression of LDL-c oxidation [18,25,60,78]. In addition, XH can increase HDL-c levels by increasing the HDL-c influx capacity and inhibiting the cholesteryl ester transfer protein (CETP) [35]. Unlike other authors, our data indicates no effect of XH on the fecal release of fats, which could be explained by the variation in animal models and doses of XH used.

Yet, the molecular mechanisms by which XH exerts its hypolipidemic effect remain unclear. Generally, AMPK is a potent hypolipidemic agent that can suppress TGs and CHOL synthesis by downregulating and phosphorylation-induced inhibiting SREBP1 and ACC-1. Indeed, AMPK downregulates and interferes with the maturation of SREBP1 and can inhibit this factor through phosphorylation of its Ser372 residue [11,15,79]. In addition, AMPK suppresses hepatic de novo lipogenesis and stimulates FA oxidation by downregulating and phosphorylation-induced suppression of the ACC-1 and ACC2 at the Ser^79^ residue and Ser^212^, respectively [15]. Interestingly, levels and activities of AMPK were significantly depleted in the livers of several obese, diabetic, and HFD-fed animal models and were associated with sustained increases in the levels and activities of SREBP1 and other lipogenic genes such as fatty acid synthase (FAS) and ACC-1 [52,56].

On the other hand, natural AMPK activators such as exercise, metformin, a-769662, hydrogen sulfide, salsalate, zingerone, quercetin, isoliquiritigenin, mangiferin, chicoric Acid, honokiol, fucoxanthin, ellagic acid, ginsenoside, AICAR, and resveratrol, prevented hepatic steatosis and NASH by AMPK-dependent downregulation and suppression of SREPB1 and its downstream target genes [11,52,56,64,80]. Similar to these data, the livers of HFD-fed rats of this study also showed a significant reduction in the levels and activities of AMPK that coincided with a significant increase in the mRNA levels of SREBP1 and ACC-1. On the other hand, and in the same page with exercise and all the above-mentioned protective drugs, XH was able to downregulate and suppress SREBP1 and ACC-1 by activating AMPK. In addition, it significantly downregulated and inhibited SREBP1 and ACC-1 in the livers of control rats, which were also parallel with a significant increase in the mRNA and phosphorylation of AMPK. These data were confirmed further after treatment with CC, which completely prevented the inhibitory effect of XH on hepatic lipid levels, as well as on the activities/levels of SREBP1 and ACC-1. Therefore, we can postulate at this stage that XH suppresses hepatic de novo lipogenesis by its exceptional ability to reduce and inhibit SREBP1 and ACC-1 by activating the AMPK signaling pathway.

Oxidative stress and inflammation are two key major events that are believed to mediate lipogenesis, liver injury, fibrosis, and the progression to NASH [5]. Keap-1/Nrf2 is the major antioxidant-signaling pathway that stimulates antioxidant expression and plays a significant role in protecting against various liver disorders and preventing NAFLD [81]. In the cell, keap-1 is the major inhibitor of Nrf2 that is tightly bound to Nrf2 in the cytoplasm to promote its degradation [81]. Under oxidative stress, keap-1 dissociates from ROS, whereas Nrf2 translocates to the nucleus to initiate the transcription cascade [81]. Conversely, sustained activation of NF-κB exaggerates oxidative stress and inflammation by promoting the production of inflammatory cytokines and adhesive molecules [82]. Like Nrf2, NF-κB is sterically maintained in the cytoplasm of resting cells by certain inhibitors known as IκB. IκB kinase (IKK) phosphorylates these inhibitors, thus disturbing the NF-κB: IκB interaction and inducing the nuclear translocation of NF-κB [82]. In addition, TNF-α and IL-6 are major inducers of NF-κB ([82]. In the livers of NAFLD, Nrf2 signaling is inhibited, whereas that of NF-κB is hyper-activated. Activating Nrf2/antioxidant signaling pathways or inhibiting the NF-κB protects against liver injury and hepatic steatosis [83,84]. In this study, we have also found that the hepatic antioxidant and anti-inflammatory protective effects of XH involve AMPK-dependent transactivation activation of Nrf2 and suppression of NF-κB. These effects were seen in the livers of the control and HFD-fed rats, suggesting a regulatory role of XH in both factors. This accounts for the higher expression of antioxidants and reduced expression of TNF-α and IL-6 in the livers of these groups of rats. In addition, XH did not alter the mRNA expression of both factors, suggesting that XH acts mainly by an AMPK-dependent mechanism that regulates the major inhibitors of both factors including keap-1 and IKK/IκB. This was not examined any further in this study and may be studied later.

Supporting our data, Wang et al. [85] have recently shown that XH alleviated T2DM-induced liver steatosis and fibrosis by mediating the NRF2/RAGE/NF-κB signaling pathway. Also, XH alleviated wound healing in diabetic rats through cysteine modification-induced inhibition of keap-1-induced Nrf2 activation and upregulation of antioxidants [45]. XH reduces inflammation and angiogenesis in pancreatic cancer cells by suppressing NF-κB [86]. Similar effects on Nrf2 and NF-κB have been shown for XH in several in vivo and in vitro studies of other animal models, such as neurodegeneration, cancer, osteoarthritis, bone loss, and hepatotoxicity [40,87,88,89]. However, AMPK signaling is closely associated with inflammation and oxidative stress via acting as an independent upstream regulator of Nrf2 and NF-κB [90,91,92,93,94,95]. These effects were independent of its metabolic effect on glucose and lipid metabolisms. Within this view, AMPK can directly activate Nrf2 by stimulating the degradation of Keap-1, phosphorylating raptor (Ser7921), and suppressing unfolded protein response (UPR) and ER stress [16,96,97]. AMPK can also inhibit the transactivation of NF-κB by phosphorylating ULK1-induced degradation of IKK [98]. AMPK can increase NAD+ levels to stimulate SIRT1, which acts as a deacetylase to inhibit NF-κB and stimulate Nrf2 [50,99,100]. Furthermore, AMPK inhibits ROS production and inflammation via downregulating NADPH oxidase and activating PGCα1 and FOXO signaling pathways [101,102,103].

## 5. Conclusions and Future Perspectives

The data obtained from this study are very interesting and the first to examine the role of hepatic AMPK activation as a possible mechanism for the hepatic protective effect of XH in a rat model of NAFLD. Our data confirmed that activation of AMPK is a major mechanism underlying the anti-obesity and anti-adiposity effect of XH in this animal model. In addition, our data support that XH-induced activation of AMPK is an indispensable major hepatic upstream mechanism responsible for the hypolipidemic, antioxidant, and anti-inflammatory effect of XH through regulating several transcription factors such as Nrf2, NF-κB, SREBP1, and ACC. However, despite this data, the study still has some limitations. Importantly, these data remain observational, and further studies using animals or cells deficit with AMPK should be carried out to confirm these mechanisms. In addition, it is well-accepted that other signaling kinases and proteins can regulate AMPK itself. These all should be targeted in future research. Furthermore, the selected dose of XH used in this study, which was based on previous evidence, was insufficient to confirm full protection against hepatic steatosis. Therefore, more studies using higher safe doses should be examined to precisely find the correct dose.

## Figures and Tables

**Figure 1 foods-12-04214-f001:**
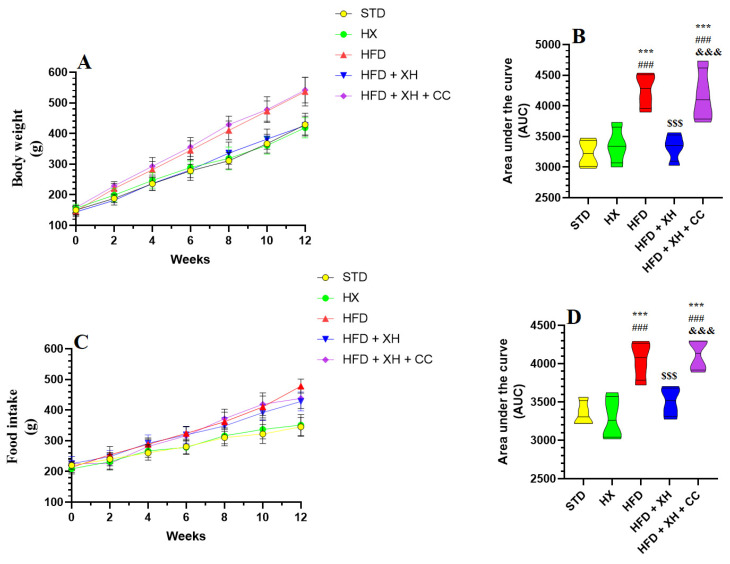
Weekly changes in food intake (**A**,**B**) and body weights (**C**,**D**) in all groups of rats. Data are presented as means ± SD for *n* = 8 rats/group. (^***^): significantly differed with STD group at *p* < 0.001; (^###^): significantly differed with XH-treated group at *p* < 0.001; (^$$$^): significantly differed with HFD group at *p* < 0.001; (^&&&^): significantly differed with HFD + XH group at *p* < 0.001. AUC: urea under the curve.

**Figure 2 foods-12-04214-f002:**
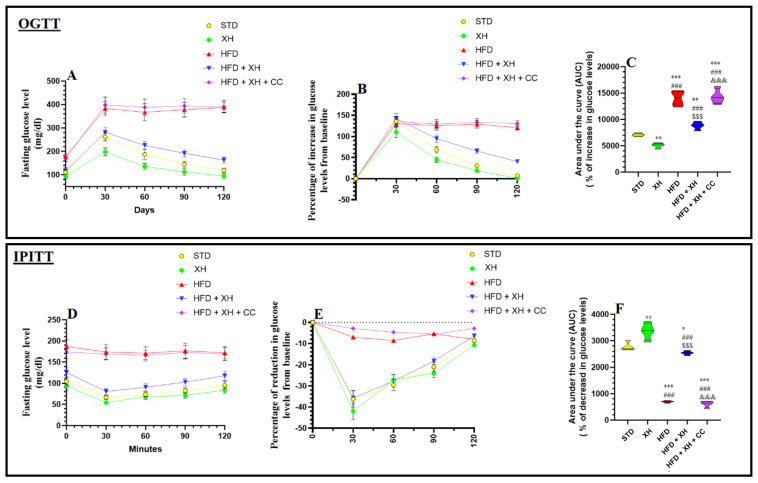
Data of oral glucose (OGTT) (**A**–**C**) and intraperitoneal insulin tolerance tests (IPITT)(**D**–**F**) in all groups of rats. Data are presented as means ± SD for *n* = 8 rats/group. (^*, **, ***^): significantly differed with STD group at *p* < 0.05, 0.01, and 0.001, respectively; (^###^): significantly differed with XH-treated group at *p* < 0.001; (^$$$^): significantly differed with HFD group at *p* < 0.001; (^&&&^): significantly differed with HFD + XH group at *p* < 0.001. UAC: urea under the curve.

**Figure 3 foods-12-04214-f003:**
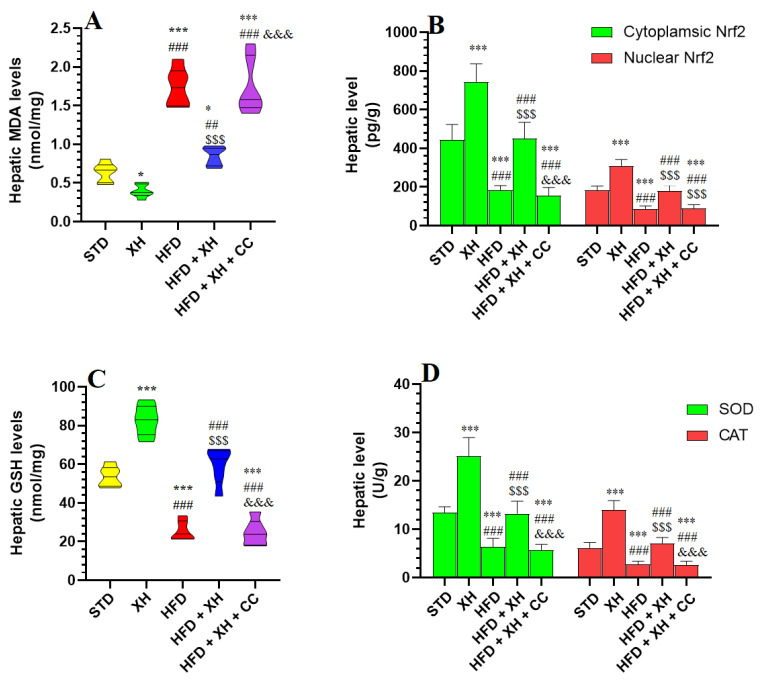
Levels of malondialdehyde (MDA) (**A**), cytoplasmic and nuclear Nrf2 (**B**), total glutathione (GSH) (**C**), superoxide dismutase (SOD), and catalase (CAT) (**D**) in the livers of all groups of rats. Data are presented as means ± SD for n = 8 rats/group. (^*, ***^): significantly differed with STD group at *p* < 0.05, 0.01, and 0.001, respectively; (^##, ###^): significantly differed with XH-treated group at *p* < 0.001; (^$$$^): significantly differed with HFD group at *p* < 0.001; (^&&&^): significantly differed with HFD + XH group at *p* < 0.001.

**Figure 4 foods-12-04214-f004:**
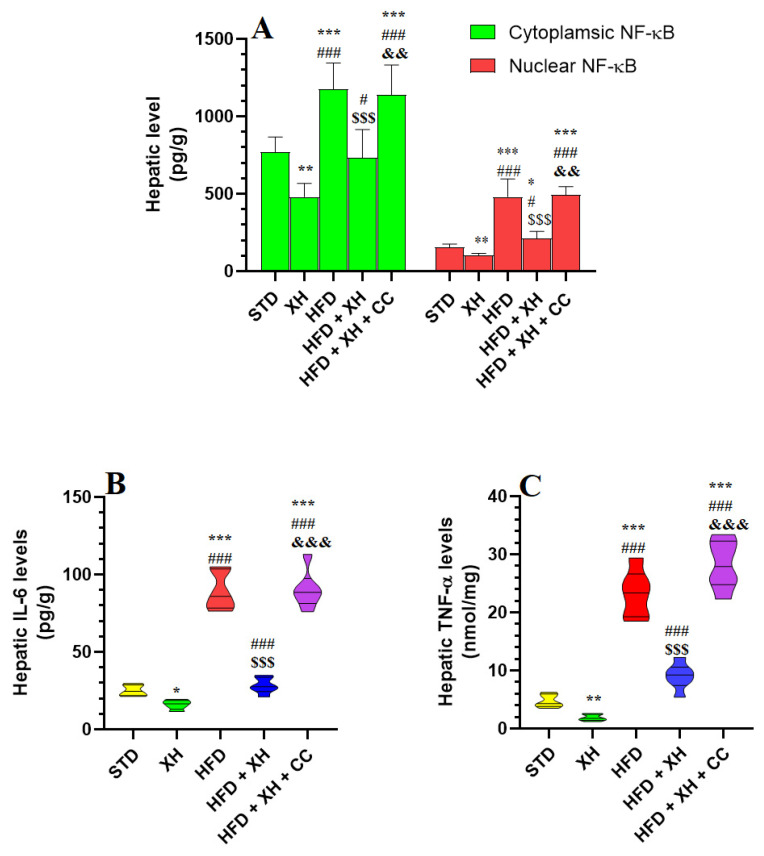
The cytoplasmic land nuclear levels of nuclear factor-kappa beta (NF-κB) (**A**), as well as levels of interleukin-6 (IL-6) (**B**) and tumor necrosis factor-α (TNF-α) (**C**) inflammation in the livers of all groups of rats. Data are presented as means ± SD for *n* = 8 rats/group. (^*, **, ***^): significantly differed with STD group at *p* < 0.05, 0.01, and 0.001, respectively; (^#, ###^): significantly differed with XH-treated group at *p* < 0.001; (^$$$^): significantly differed with HFD group at *p* < 0.001; (^&&, &&&^): significantly differed with HFD + XH group at *p* < 0.001.

**Figure 5 foods-12-04214-f005:**
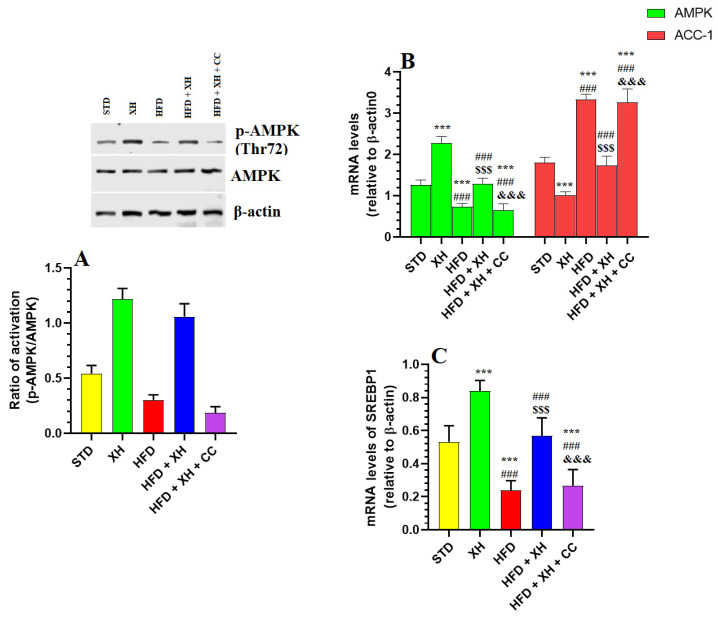
Protein expression of p-AMPK/AMPK (**A**), as well as mRNA expression of AMPK, SREBP1, and ACC-1 in the livers of HFD-fed rats (**B**,**C**). Data are presented as means ± SD for *n* = 4 rats/group. (^***^): significantly differed with STD group at *p* < 0.01 and 0.001, respectively; (^###^): significantly differed with XH-treated group at *p* < 0.001; (^$$$^): significantly differed with HFD group at *p* < 0.001; (^&&&^): significantly differed with HFD + XH group at *p* < 0.001. AMPK: AMPK-activated protein kinase and ACC-1: acetyl-CoA carboxylase.

**Figure 6 foods-12-04214-f006:**
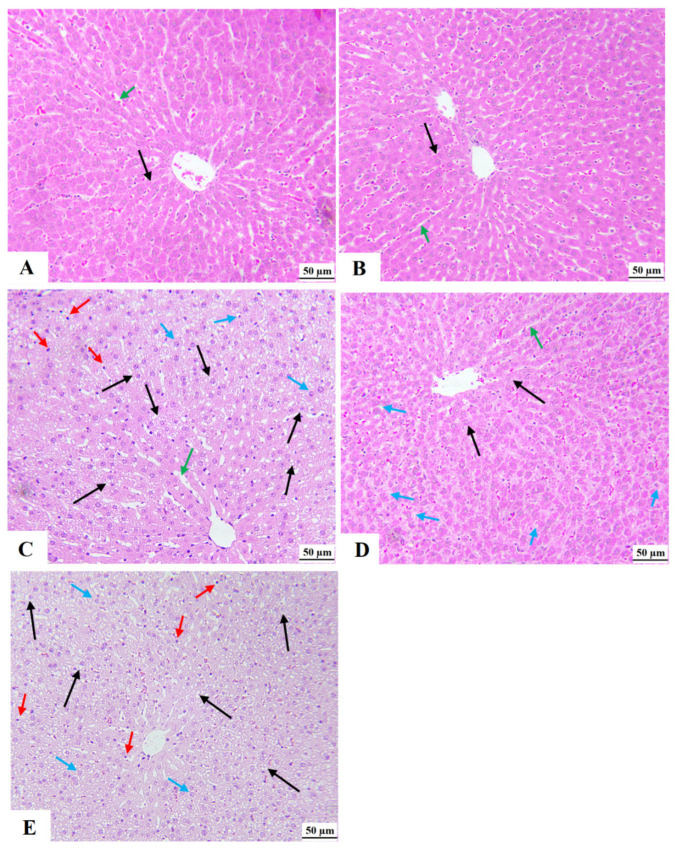
Morphological images for the livers of all groups of rats as stained by hematoxylin and eosin stain (magnification = 200×). (**A**,**B**): were taken from control rat and XH-treated rats, respectively, and showed normal histological features, including central vein, hepatocyte (black arrows), and sinusoids (green arrows). (**C**): Represents HFD-fed rats and showed a massive increase in the number of cytoplasmic fat granules/vacuoles in the majority of the hepatocytes (black arrows). These livers also showed dilated sinusoids (green arrows) and an increased number of infiltrating immune cells (red arrows) and pyknotic cells (blue arrows). (**D**): was taken from HFD + XH-treated rats and showed much improvement in the structure of hepatocytes (black arrow) and normally sized- sinusoids. However, fat vacuoles were still seen in the hepatocytes that are away from the central vein but in smaller quantities as compared to HFD-fed rats in (**C**). (**E**): was taken from HFD + XH + CC and showed almost similar pathological changes to those seen in the HFD-fed rats.

**Figure 7 foods-12-04214-f007:**
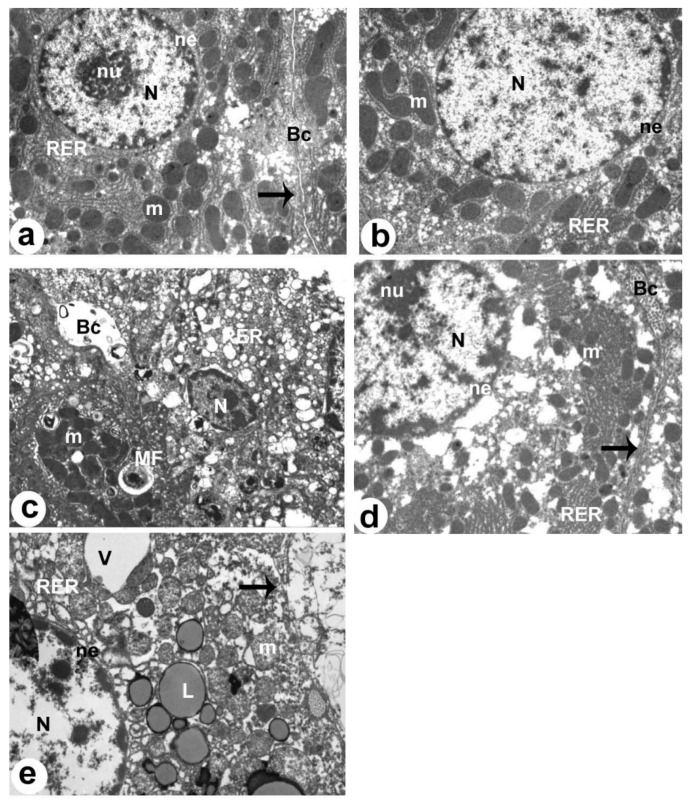
Electron micrographs of the livers of all groups of rats as taken by electron microscopy. (**a**,**b**): were taken from control and XH-treated rats. Both livers showed a normal hepatocyte (H) with an intact nucleus (N), rough endoplasmic reticulum (RER), and mitochondria (m). The normally appeared nucleus was surrounded by an intact nuclear envelope (NE) and contained nucleolus (nu) and clear chromatin masses (Chr). The hepatocyte plasma membrane (arrow) was also seen and intact. Normal bile canaliculus (Bc) with intact microvilli were also seen. (**c**): was taken from NAFLD-treated rats and showed obvious apoptotic and degenerated hepatocytes with pyknotic nucleus (N), increased amounts of lipids (L), and myelin figures (MF). The RER was damaged and dilated. Dilated bile canaliculus (Bc) with damaged microvilli were also seen. (**d**): was taken from an HFD + XH-treated rat and showed improvement in the structure of the hepatocyte (H) and bile canaliculus (Bc) with partial changes in some mitochondria (m) and the RER. (**e**): was taken from HFD + XH + CC-treated rats and showed similar ultrastructural changes to those that appeared in the livers of HFD-fed rats with larger lipid droplets (L) and vacuoles (V).

**Figure 8 foods-12-04214-f008:**
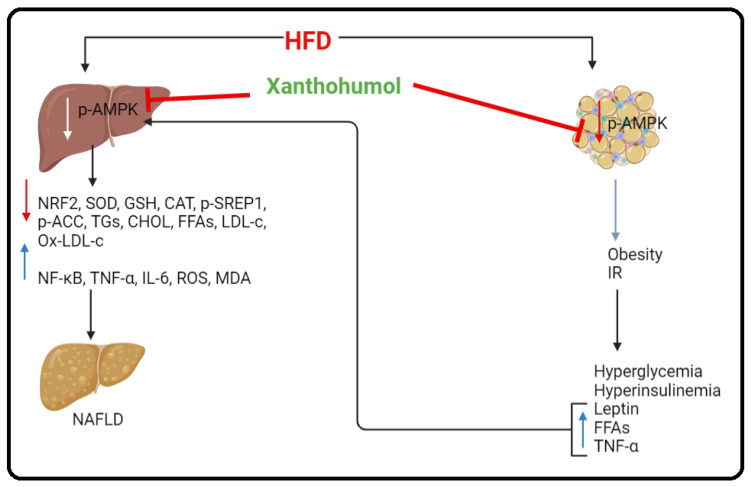
A schematic diagram showing the AMPK-dependent mechanism by which Xanthohumol alleviated high-fat diet (HFD)-induced non-alcoholic fatty liver disease in rats.

**Table 1 foods-12-04214-t001:** Primers used in the qPCR reaction.

Target	Primers Sequence 5′→3′	Accession No.	Base Pair Length
AMPK	F: GAAGTCAAAGCCGACCCAAT R: AGGGTTCTTCCTTCGCACAC	NM_019142	116
SREBP1c	F: GCTCACAAAAGCAAATCACT R: GCGTTTCTACCACTTCAGG	NM_001276707.1	191
ACC-1	F: TGAGGAGGACCGCATTTATC R: AAGCTTCCTTCGTGACCAGA	NM_022193.1	221
β-actin	F: CGAGTACAACCTTCTTGCAGC R: CCTTCTGACCCATACCCACC	NM_031144.3	209

**Table 2 foods-12-04214-t002:** Adiposity markers in all groups of rats.

Parameter	STD	XH	HFD	HFD + XH	HFD + XH + CC
Final body weight (g)	429.3 ± 37.3	419.8 ± 33.5	537.4 ± 47.4 ***^,###^	423.6 ± 31.9 ^$$$^	524.4 ± 51.7 ***^,###,&&&^
Subcutaneous fat (g)	4.1 ± 0.3	4.4 ± 0.5	8.2 ± 0.7 ***^,###^	4.8 ± 0.4 *^,$$$^	7.9 ± 0.8 ***^,###,&&&^
Epididymal fat (g)	6.8 ± 0.6	6.1 ± 0.7	13.5 ± 1.5 ***^,###^	8.3 ± 0.8 *^,#,$$$^	14.1 ± 1.3 ***^,###,&&&^
Peritoneal fat (g)	3.2 ± 0.2	3.3 ± 0.4	7.6 ± 0.5 ***^,###^	4.3 ± 0.3 *^,#,$$$^	7.1 ± 0.8 ***^,###,&&&^
Fasting glucose (mg/dL)	111.4 ± 9.7	94.2 ± 8.6 **	178.1 ± 16.4 ***^,###^	116.3 ± 10.1 ^#,$$$^	169.8 ± 11.4 ***^,###,&&&^
Fasting insulin (µIU/mL)	4.3 ± 0.6	4.1 ± 0.4	7.9 ± 0.7 ***^,###^	5.1 ± 0.5 **^,#,$$$^	8.2 ± 0.7 ***^,###,&&&^
HOMA-IR	1.17 ± 0.1	0.91 ± 0.08 *	3.2 ± 0.4 ***^,###^	1.5 ± 0.2 **^,###,$$$^	3.4 ± 0.4 ***^,###,&&&^
Serum leptin (ng/mL)	34.2 ± 3.4	36.5 ± 3.1	94.5 ± 8.3 ***^,###^	46.7 ± 5.1 **^,##,$$$^	89.5± 9.4 ***^,###,&&&^
Serum FFAs (mmol/L)	1.2 ± 0.1	0.84 ± 0.05 *	2.7 ± 0.2 ***^,###^	1.4 ± 0.1 *^,###,$$$^	2.5 ± 0.3 ***^,###,&&&^
Serum TNF-α (pg/mL)	265.2 ± 19.2	187.2 ± 15.7 *	655.3 ± 57.4 ***^,###^	349.1 ± 28.3 *^,###,$$$^	638.9 ± 68.3 ***^,###,&&&^

Data are presented as means ± SD for *n* = 8 rats/group. (^*, **, ***^): significantly differed with STD group at *p* < 0.05, 0.01, and 0.001, respectively; (^#^, ^##^, ^###)^: significantly differed with XH-treated group at *p* < 0.05, 0.01, and 0.001, respectively; (^$$$^): significantly differed with HFD group at *p* < 0.001; (^&&&^): significantly differed with HFD + XH group at *p* < 0.001. TNF-α: tumor necrosis-factor-α; FFAs: free fatty acids; and HOMA-IR: the homeostasis model of insulin resistance index.

**Table 3 foods-12-04214-t003:** Serum, hepatic, and fecal lipid levels in all groups of rats.

	Parameter	STD	XH	HFD	HFD + XH	HFD + XH + CC
Serum	TGs (mg/dL)	89.3 ± 9.1	77.3 ± 6.5 **	168.3 ± 15.2 ***^,###^	95.3 ± 8.6 ^###,$$$^	175 ± 17.8 ***^,###,&&&^
	CHOL (mg/dL)	75.6 ± 6.9	61.3 ± 5.7 **	187.4 ± 16.4 ***^,###^	96.4 ± 7.4 **^,###,$$$^	194.3 ± 17.3 ***^,###,&&&^
	HDL-c (mg/dL)	23.2 ± 2.1	26.7 ± 2.4	11.5 ± 1.9 ***^,###^	24.5 ± 2.7 ^$$$^	10.8 ± 9.7 ***^,###,&&&^
	LDL-c (mg/dL)	32.3 ± 3.6	21.5 ± 2.6 *	99.6 ± 8.5 ***^,###^	41.4 ± 3.9 *^,###,$$$^	103.1 ± 9.1 ***^,###,&&&^
	Ox-LDL-c (ng/mL)	11.3 ± 1.1	6.8 ± 0.71 ***	44.8 ± 3.6 ***^,###^	17.8 ± 2.5 **^,###,$$$^	48.9 ± 5.1 ***^,###,&&&^
Liver	TGs (ng/g tissue)	225.4 ± 19.5	148.2 ± 11.4 **	564.2 ± 42.2 ***^,###^	258.2 ± 21.4 *^,###,$$$^	610.3± 54.6 ***^,###,&&&^
	CHOL (ng/g tissue)	78.4 ± 5.7	58.7 ± 4.7 **	198.4 ± 15.4 ***^,###^	85.7 ± 8.9 **^,###,$$$^	184.4 ± 16.7 ***^,###,&&&^
	FFAs (μmol/g tissue)	125.4 ± 11.4	89.5 ± 6.9 ***	349.5 ± 46.7 ***^,###^	167.3 ± 17.8 *^,###,$$$^	324 ± 33.1 ***^,###,&&&^
Stool	TGs (ng/g)	2.1 ± 0.3	2.4 ± 0.4	4.9 ± 0.4 ***^,###^	4.4 ± 0.6 ***^,###^	5.3 ± 0.5 ***^,###^
	CHOL (ng/g)	2.6 ± 0.2	2.5 ± 0.3 **	6.3 ± 0.5 ***^,###^	6.1 ± 0.6 ***^,###^	5.9 ± 0.6 ***^,###^

Data are presented as means ± SD for n = 8 rats/group. (^*, **, ***^): significantly differed with STD group at *p* < 0.05, 0.01, and 0.001, respectively; (^###^): significantly differed with XH-treated group at *p* < 0.001; (^$$$^): significantly differed with HFD group at *p* < 0.001; (^&&&^): significantly differed with HFD + XH group at *p* < 0.001. TGs: triglycerides; CHOL: cholesterol; HDL-c: high-density lipoprotein cholesterol; LDL-c: low-density lipoprotein cholesterol; ox-LDL: oxidized LDL; and FFAs: free fatty acids.

**Table 4 foods-12-04214-t004:** Serum levels of liver function enzymes in all groups of rats.

Parameter	STD	XH	HFD	HFD + XH	HFD + XH + CC
ALT (U/L)	27.8 ± 2.6	25.8 ± 3.5	76.4 ± 5.3 ***^,###^	35.2 ± 3.1 ***^,###,$$$^	81.4 ± 8.3 ***^,###,&&&^
AST (U/L)	16.6 ± 1.2	19.3 ± 3.2	88.7 ± 6.5 ***^,###^	27.8 ± 2.8 **^,###,$$$^	86.1 ± 6.9 ***^,###,&&&^
GGT (U/L)	21.5 ± 2.9	24.3 ± 2.6	105.3 ± 8.9 ***^,###^	38.2 ± 3 ***^,###,$$$^	99.2 ± 10.6 ***^,###,&&&^

Data are presented as means ± SD for *n* = 8 rats/group. (^**, ***^): significantly differed with STD group at *p* < 0.05, 0.01, and 0.001, respectively; (^###^): significantly differed with XH-treated group at *p* < 0.001; (^$$$^): significantly differed with HFD group at *p* < 0.001; (^&&&)^: significantly differed with HFD + XH group at *p* < 0.001. AST: aspartate aminotransferase (AST); ALT: Alanine aminotransferase; and GTT: gamma-glutamyl transpeptidase.

## Data Availability

The datasets used and analyzed in the current study are available from the corresponding author upon reasonable request.

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
