# Peer review of "The Hepatic Antisteatosis Effect of Xanthohumol in High-Fat Diet-Fed Rats Entails Activation of AMPK as a Possible Protective Mechanism"

_foods, 2023, doi:10.3390/foods12234214_

Round 1

Reviewer 1 Report

Comments and Suggestions for Authors

The study described focuses on investigating the potential anti-steatosis effect of Xanthohumol (XH) in high-fat diet (HFD)-fed rats, with a particular emphasis on its interaction with AMPK (adenosine monophosphate-activated protein kinase). In summary, this study demonstrates that Xanthohumol (XH) exhibits anti-steatosis effects in HFD-fed rats by improving various metabolic parameters, reducing liver fat accumulation, and exerting antioxidant and anti-inflammatory effects. The activation of AMPK appears to play a crucial role in mediating these effects, as inhibiting AMPK reversed the benefits of XH. This research suggests the potential of XH as a dietary supplement or pharmaceutical intervention for managing obesity-related liver steatosis and associated metabolic dysregulation.

While this study is interesting, there are several suggestions to enhance the quality of the document:

Introduction:

The authors could provide more comprehensive information about Xanthohumol (XH) to offer a better understanding of its background and significance.

Materials and Methods:

The authors should correct an error in the Materials and Methods section, where it stated that GAPDH was used as the housekeeping gene, and replace it with "β-actin," which was the actual gene used.

The authors should specify whether they calculated the G-power value to ensure the study's statistical power and whether the number of animals used aligns with recommended sample size considerations.

Citation:

It is advisable to add a citation to the work by Folch et al. (1957) for proper referencing.

Figure 6:

The size and clarity of Figure 6 should be improved, particularly in the 6c portion. Consider using Oil Red O stain to enhance the visibility of lipid droplets in the figure. You can find some protocols in this papers:

https://doi.org/10.3390/ijms232012151

Discussion:

In the Discussion section, the authors should correct "XU" to "XH."

To confirm the effect of AMPK on browning, it is suggested to check the expression of UCP1.

The sentence "Yet, the molecular mechanisms by which XH exerts its hypolipidemic effect are still not unclear" should be corrected to "However, the molecular mechanisms by which XH exerts its hypolipidemic effect remain unclear."

Author Response

Comments from reviewer 1:

While this study is interesting, there are several suggestions to enhance the quality of the document:

General response:

Dear reviewer, first of all, all authors would like to thank you for the time and effort given to review this manuscript. All your comments are correct and placed to improve the quality of this paper as you have indicated above. We have corrected most of your comments and if not, we have provided an explanation. Your comment improved the presentation of this manuscript and we hope it is now acceptable for publication. Please find our response point by point below and all changes corresponding to your comments are indicated by yellow colors in the text.

Comment 1: Introduction:

The authors could provide more comprehensive information about Xanthohumol (XH) to offer a better understanding of its background and significance.

Response: Thank you for this comment. This was so valuable. We have extended our introductory part concerning HX effect by describing its metabolic and multi-organ effects which are related to the target of this manuscript and the aim of the study. We have enriched the introduction with sufficient background about HX to introduce the reader to the topic and make the paper more readable.

Comment 2: Materials and Methods: The authors should correct an error in the Materials and Methods section, where it stated that GAPDH was used as the housekeeping gene, and replace it with "β-actin," which was the actual gene used.

Response: This is a printing mistake, The actual gene as indicated by the primers in table 1 and in the results and figures is actin, we have corrected this.

Comment 3: The authors should specify whether they calculated the G-power value to ensure the study's statistical power and whether the number of animals used aligns with recommended sample size considerations.

Response: We understand the point of the reviewer. In animal research, 6-8 samples is enough for the study as the model is validated. G-power is preferred used with population studies. However, we have confirmed the normal distribution of our samples and conducted the analysis as described. This is an appropriate and correct way for data analysis in this type of research.

Comment 4: Citation: It is advisable to add a citation to the work by Folch et al. (1957) for proper referencing.

Response: This reference was cited as requested.

Comment 6: Figure 6: The size and clarity of Figure 6 should be improved, particularly in the 6c portion. Consider using Oil Red O stain to enhance the visibility of lipid droplets in the figure. You can find some protocols in this papers:

Response: the quality of this figure is high and with publication quality. This confusion about the clarity of the figure is due to the small figure size as placed in the manuscript word document. The reviewer can magnify the figure in the doc and will see the high quality of all images. We think on publications, the publisher will use large space for these images and they will be better presented as usual. We have experience with this type of publication where the figures in the submitted file (included in the text) appear not clear but on the published version are excellent.

With respect to oil-red, all authors agree with the reviewer that they make fat vacuoles appear in a good way and reflect fat deposition. This protocol requires a fresh frozen sample, which we don’t have at this stage. As per the reviewer's recommendation, we will consider this in future research as we have applied for external funds to continue our search on HX.

Comment 7: Discussion: In the Discussion section, the authors should correct "XU" to "XH."

Response: corrected

Comment 8: To confirm the effect of AMPK on browning, it is suggested to check the expression of UCP1.

Response: Correct. However, two major reasons prevent us from performing this at this stage. First of all, is the lack of extra funds, and second is the long time needed for the request and arrival of these antibodies or primers of this gene. As we explained in the previous comment, we have applied for external funds and waiting for approval. Therefore, our research on the mechanisms by which HX protects against NAFLD and obesity will continue, and we will consider all missing points.

Comment 9:  The sentence "Yet, the molecular mechanisms by which XH exerts its hypolipidemic effect are still not unclear" should be corrected to "However, the molecular mechanisms by which XH exerts its hypolipidemic effect remain unclear."

Response: Correct as needed.

Reviewer 2 Report

Comments and Suggestions for Authors

This is an interesting report expanding knowledge on the potential therapeutic effects of Xanthohumol against metabolic syndrome-associated disorder, specifically hepatic steatosis.

The introduction provides a strong/relevant background for the purpose of the study, while the methodology is also descriptive enough. The results are clear, although the experiments lack the comparative control, especially the use of metformin as a control drug could have enhanced the value of the report.

Specific comments

Define all abbreviations at first mention, including DMSO.

Give details of diet, including fat composition, if relevant

Add the reference by “Folch et al. (1957)”

Give details of the microscope used, for that matter for all equipment used.

Avoid using repeated words in one sentence, for example, the use of “significantly” in one sentence should be avoided”

Remove the words “…in an AMPK-Dependent Manner” in your subheadings within the results section, and only add this phrase if relevant, or at least where you are reporting on the effects of treatment in regulating “AMPK”

Figure captions should describe all abbreviations used within that particular figure.

If you are stating “Previous studies…” then relevant references should be added

I agree the results suggest the AMPK mechanism by which Xanthohumol exerts its therapeutic effects. However, I feel authors should clearly highlight this a “potential” or “possible” mechanism of action rather than appear as a definitive mechanism. Edit the manuscript to reflect this.

Author Response

Comments from reviewer 2: This is an interesting report expanding knowledge on the potential therapeutic effects of Xanthohumol against metabolic syndrome-associated disorders, specifically hepatic steatosis. The introduction provides a strong/relevant background for the purpose of the study, while the methodology is also descriptive enough. The results are clear, although the experiments lack comparative control, especially the use of metformin as a control drug could have enhanced the value of the report. General response: General response: Dear reviewer, first of all, all authors would like to thank you for the time and effort given to review this manuscript. All your comments are correct and placed to improve the quality of this paper as you have indicated above. We have corrected all your comments as requested. Please find our response point by point below and all changes corresponding to your comments are indicated by yellow in the text. We hope the manuscript is now acceptable for publication. Comment 1: Define all abbreviations at first mention, including DMSO. Response: we have revised the text and made sure that all abbreviations are defined as they first appear. For example triglycerides and cholesterol in the introduction and they corrected all through the next text. DMSO was identified. Comment 2: Give details of diet, including fat composition, if relevant Response: we have added the carb, protein, and fat ratios of each diet. HFD was based on lard and we have added the percentages and energy obtained from lard to each diet. If the reviewer still needs it, we can add a table for the ingredients of all these diets. But as we said the reader can get them on the company website and this will reduce the number of tables in this paper. Comment 2: Add the reference by “Folch et al. (1957)” Response: This was added Comment 3: Give details of the microscope used, for that matter for all equipment used. Response: this was added and the same approachh was followed for any other further instrument Comment 4: Avoid using repeated words in one sentence, for example, the use of “significantly” in one sentence should be avoided” Response: Thank you for this comment. We have corrected this in multiple positions. Comment 5: Remove the words “…in an AMPK-Dependent Manner” in your subheadings within the results section, and only add this phrase if relevant, or at least where you are reporting on the effects of treatment in regulating “AMPK” Response: This has been corrected as suggested. Comment 6: Figure captions should describe all abbreviations used within that particular figure. Response: All abbreviations in any figure or table were mentioned as full names in the caption Comment 7: Comment If you are stating “Previous studies…” then relevant references should be added Response: correct. We made sure to reference all these sentences as suggested. Comment 8: I agree the results suggest the AMPK mechanism by which Xanthohumol exerts its therapeutic effects. However, I feel authors should clearly highlight this a “potential” or “possible” mechanism of action rather than appear as a definitive mechanism. Edit the manuscript to reflect this. Response: We believe the reviewer suggested altering the manuscript title. Please confirm. However, the current title indicates that the protective effect of HX entails activation of AMPK as a possible, not definite mechanism.

Reviewer 3 Report

Comments and Suggestions for Authors

Regarding the manuscript entitled "The Hepatic Antisteatosis Effect of xanthohumol in High-Fat Diet-Fed Rats Entails Activation of AMPK"

The study is good and scientifically sound and needs some major revisions, please find my comments listed below:

1-The abstract needs addition of short introductory paragraph on you study merits.

2-The introduction section needs to be shorter and reduce the number of citations "54 reference in the introduction why???"

3-Introduction contains several typo mistakes.

4-Materials and methods needs some revisions for the name of the plant and typo mistakes "see the attached file"

5-Results

-What is the difference between subtitle 3.1 and 3.2????

-The abbreviation of Xanthohumol in text, figures, ant tables some times XH and other HX please unify.

-full of typo mistakes please revise.

 6- Discussion needs to be shorter and focused on the major findings of the study and please reduce the number of citations.

7-Please add a separate conclusion section with the major finding and future recommendations.

Finally, the study is well designed and scientifically sound and suitable for foods after the suggested revisions.

Comments on the Quality of English Language

The manuscript is full of typo-errors and need extensive English revisions.

Author Response

Comments from reviewer 3:

Regarding the manuscript entitled "The Hepatic Antisteatosis Effect of xanthohumol in High-Fat Diet-Fed Rats Entails Activation of AMPK. The study is good and scientifically sound and needs some major revisions, please find my comments listed below:

Comment 1: The abstract needs the addition of a short introductory paragraph on your study merits

Response: we have added a short introduction to the abstract.

2-The introduction section needs to be shorter and reduce the number of citations "54 references in the introduction why???"

Response: The introduction is written in a comprehensive way to link obesity with the pathogenesis of NAFLD. It also describes the alterations in AMPK, NF-kB, and NRf2 and how these factors are linked with each other in the development and progression of NAFLD. It also described the pharmacological effects of XH and its effect on adiposity, hyperglycemia, hyperlipidemia, and NALFD. Furthermore, it describes previous studies that have shown the regulation of AMPK, Nrf2, and NF-kB by HX. All these supporting studies are needed in the introduction to tell our story and to clarify the major objective of this study. The text of the introduction contains relevant information and some other reviewers suggested adding more text for the pharmacological effect of HX. However, based on the comments and suggestions of the reviewer, we have deleted some other references and kept only those closely related to each sentence.  

Comment 3: The introduction contains several typo mistakes.

Response: all mistakes in the English language, grammar, and pronunciation were corrected in the whole manuscript.

Comment 4: Materials and methods need some revisions for the name of the plant and typo mistakes "see the attached file"

Response: All mistakes in the attached pdf file were considered and corrected. The spelling in HX and XH were corrected to XH in the text and figures.

Comment 5: What is the difference between subtitles 3.1 and 3.2????

Response: this was corrected

Comment 6: The abbreviation of Xanthohumol in text, figures, and tables sometimes XH and other HX please unify.

Response: This was corrected in the text, tables, and figures

Comment 7: full of typo mistakes please revise.

Response: all mistakes were corrected.

Comment 8 Please add a separate conclusion section with the major findings and future recommendations.

Response: This was added.

Round 2

Reviewer 1 Report

Comments and Suggestions for Authors

thank you for your replies